# A Comparative Study on the Neuroprotective Effect of Geopung-Chunghyuldan on In Vitro Oxygen–Glucose Deprivation and In Vivo Permanent Middle Cerebral Artery Occlusion Models

**DOI:** 10.3390/ph16040596

**Published:** 2023-04-15

**Authors:** Tae-Hoon Park, Han-Gyul Lee, Seung-Yeon Cho, Seong-Uk Park, Woo-Sang Jung, Jung-Mi Park, Chang-Nam Ko, Ki-Ho Cho, Seungwon Kwon, Sang-Kwan Moon

**Affiliations:** 1Department of Korean Medicine Cardiology and Neurology, Graduate School, Kyung Hee University, Seoul 02447, Republic of Korea; 2Department of Cardiology and Neurology, College of Korean Medicine, Kyung Hee University, Seoul 02447, Republic of Korea

**Keywords:** Geopung-Chunghyuldan, Chunghyuldan, neuroprotection, oxygen–glucose deprivation, permanent middle cerebral artery occlusion

## Abstract

Geopung-Chunghyuldan (GCD), which is a mixture of Chunghyuldan (CD), Radix Salviae Miltiorrhizae, Radix Notoginseng, and Borneolum Syntheticum, is used to treat ischemic stroke in traditional Korean medicine. This study aimed to investigate the effects of GCD and CD on ischemic brain damage using in vitro and in vivo stroke models, as well as to elucidate the synergistic effects of GCD against ischemic insult. To study the effect of GCD in an in vitro ischemia model, SH-SY5Y cells were exposed to oxygen–glucose deprivation (OGD). Cell death after 16 h of OGD exposure was measured using the MTT assay and live/dead cell counting methods. An in vivo ischemia mice model was established through permanent middle cerebral artery occlusion (pMCAO). To determine the neuroprotective effect of GCD, it was orally administered immediately and 2 h after pMCAO. The infarct volume was measured through 2,3,5-triphenyltetrazolium chloride staining at 24 h after pMCAO. Compared with the control group, GCD treatment significantly reduced OGD-induced cell death in SH-SY5Y cells; however, CD treatment did not show a significant protective effect. In the pMCAO model, compared with the control group, treatment with GCD and CD significantly and mildly reduced the infarct volume, respectively. Our findings indicate that compared with CD, GCD may allow a more enhanced neuroprotective effect in acute ischemic stroke, indicating a potential synergistic neuroprotective effect. The possibility of GCD as a novel alternative choice for the prevention and treatment of ischemic stroke is suggested.

## 1. Introduction

Stroke is the third and fourth leading cause of annual deaths in Korea [1] and the United States [2,3], respectively. Worldwide, stroke is the leading cause of permanent disability in adults [3,4]. Approximately 90% of stroke cases are cerebral infarctions caused by thrombus or embolization [2,3,5,6]. Further, its incidence is increasing, given the increasingly aging society; accordingly, urgent measures are required from a health–economic perspective.

Tissue plasminogen activator (tPA), which is a thrombolytic agent, is currently the only drug approved by the U.S. Food and Drug Administration for treating cerebral infarction, and should be administered within 3 h of the onset of acute cerebral infarction [7]. However, tPA is clinically used only in limited cases (4–7%) of patients with acute cerebral infarction, given the short therapeutic window, risk of bleeding, and nonresponsiveness in some patients [3,8]. Numerous studies have attempted to develop neuroprotection drugs for cerebral infarction; however, there have been no successful clinical trials [9]. To overcome these challenges, there are currently many studies on new treatment methods being conducted [3,6,9].

There has been increasing attention on traditional Korean medicine, which applies natural products, as a complementary or alternative treatment for cerebral infarction [10,11,12,13]. Recently, Kim et al. [14] developed a drug comprising Coptidis Rhizoma, Scutellariae Radix, Phellodendri Cortex, Gardeniae Fructus, and Rhei Radix et Rhizoma extracted with 80% ethanol and named it Chunghyuldan (CD), which has been extensively studied [13,14,15,16,17,18,19,20,21]. Specifically, a study using an animal model of cerebral infarction demonstrated the neuroprotective effect of CD [21]. Additionally, a clinical study on patients with small-vessel cerebral infarction showed that CD exerted a significant inhibitory effect on cerebral infarction recurrence, compared with antiplatelet treatment [13,22,23].

Numerous studies have investigated the treatment of ischemic heart disease using cardiotonic pills, which comprise Salviae Miltiorrhizae Radix, Notoginseng Radix et Rhizoma, and Bomeolum [24,25,26]; further, cardiotonic pills have been shown to exert neuroprotective effects [27,28,29]. Based on this evidence, in anticipation of the enhancement of the neuroprotective effect when CD and cardiotonic pills are used together, Kyung Hee University Korean Medicine Hospital Stroke and Brain Diseases Center developed Geopung-Chunghyuldan (GCD) by combining CD with cardiotonic pills. However, the neuroprotective effect of GCD and the difference in efficacy from CD have not been elucidated, to date. Therefore, studies should investigate whether combining CD and the herbal material used in cardiotonic pills could exert a synergistic effect that could enhance the therapeutic efficacy against cerebral infarction.

We aimed to evaluate the neuroprotective effect of Geopung-Chunghyuldan (GCD) combined with Salviae Miltiorrhizae Radix, Notoginseng Radix et Rhizoma, and Bomeolum using in vitro and in vivo stroke models.

## 2. Results

### 2.1. HPLC Analysis Results

#### 2.1.1. Quantitative Analysis of Salvianolic Acid B and Tanshinone II A

We performed HPLC analysis for quantitative analysis of the Salviae Miltiorrhizae Radix and Notoginseng Radix et Rhizoma, which constituted cardiotonic pills and Sample B. Figure 1 shows the HPLC chromatogram of salvianolic acid B and tanshinone II A, which are components of Salviae Miltiorrhizae Radix, in cardiotonic pills and Sample B. Cardiotonic pills contained 0.14% of salvianolic acid B and 0.002% of tanshinone II A, while the corresponding values in Sample B were 2.66% and 0.11%, respectively.

#### 2.1.2. Quantitative Analysis of Ginsenoside Rg1 and Rb1

Figure 2 shows the HPLC chromatogram of ginsenoside Rg1 and Rb1, which are components of Notoginseng Radix et Rhizoma, in cardiotonic pills and Sample B. Cardiotonic pills contained 0.36% of ginsenoside Rg1 and 0.34% of Rb1, while Sample B contained 0.83% of ginsenoside Rg1 and 0.95% of Rb1.

### 2.2. Neuroprotective Effects of the Experimental Drugs in the OGD Model

#### Determination of the Effect of Treatment with Experimental Drugs on Cell Death

We examined whether treatment with the experimental drugs affected cell viability. After 48 h of treatment with the experimental drugs, experimental drug treatment did not significantly alter cell viability in both the MTT assay and cell count experiments (Table 1 and Table 2, Figure 3 and Figure 4).

### 2.3. Neuroprotective Effect of Experimental Drugs against OGD-Induced Cell Death

#### 2.3.1. MTT Assay Results

OGD exposure significantly decreased the optical density (O.D.) value of the cells. Compared with the control group, Sample A + B + C complex (both concentrations of A10 + B1 + C0.1 (μg/mL) and A50 + B5 + C0.5 (μg/mL)) significantly increased the O.D. value. On the other hand, Sample A and Sample B + C showed an increase in O.D. value compared to control in all combinations, but there was no statistical significance (Table 3, Figure 5).

#### 2.3.2. Cell Count Results

The cell count experiment showed that OGD exposure significantly reduced the number of live cells compared with the group not treated with OGD. Further, compared with OGD-treated control groups, Sample A + B + C complex (A50 + B5 + C0.5 [μg/mL]) significantly increased the cell viability compared with the control group. Contrastingly, Sample A and Sample B + C showed an increase in cell viability compared to control, but there was no statistical significance in all combinations (Table 4, Figure 6).

#### 2.3.3. Live/Dead Viability Cytotoxicity Kit Experiment Results

OGD exposure resulted in cell death, which was significantly reduced by treatment with the experimental drugs (Figure 7).

### 2.4. Neuroprotective Effect of Experimental Drugs in the pMCAO Model

Compared with the control group, treatment with Sample A (45 mg/kg) nonsignificantly decreased the cerebral infarct volume (Table 5, Figure 8), while Sample B + C (mixing ratio, 22.5:1.2 mg/Kg) significantly decreased the cerebral infarct volume (Table 6, Figure 9). Compared with the control group, the Sample A + B + C complex extract (fixed concentration of Sample A, all combinations) nonsignificantly decreased the cerebral infarct volume (Table 7, Figure 10). Moreover, Sample A + B + C complex extract (fixed Sample B + C concentration) reduced the cerebral infarct volume; specifically, there was a significant decrease in the cerebral infarct volume at the mixing ratio of 45:7.5:1.2 (mg/kg) (Table 8, Figure 11).

## 3. Discussion

This study evaluated the neuroprotective effect of GCD in acute cerebral infarction using both in vitro and in vivo models. We found that GCD showed a significant neuroprotective effect in both models, indicating an improved synergistic neuroprotective effect compared with conventional CD.

Experimental models for studying cerebral infarction can be mainly divided into in vitro and animal models [6]. In vitro models typically apply glucose deprivation (GD) or OGD. The GD method involves neuronal culturing in a glucose-free medium. Contrastingly, the OGD method combines the GD method with hypoxic exposure and is widely used, since it is relatively more similar to the method for in vivo cerebral infarction [6,30,31,32]. Animal models for cerebral infarction are mainly classified into the global and focal ischemic stroke models. The global ischemic stroke model can either involve four-vessel or two-vessel occlusion, depending on the occlusive blood vessel; further, it clinically mimics hypoxic brain injury due to cardiac arrest. The focal ischemic stroke model involves transient, permanent, or photothrombotic distal middle cerebral artery occlusion; further, it mimics cerebral infarction in humans. The transient method, which temporarily occludes and recanalizes the middle cerebral artery, and the pMCAO method, which permanently occludes the middle cerebral artery by cauterizing or binding with a thread, are widely used depending on the study purpose [6,31,33,34,35]. The present study used the aforementioned OGD and pMCAO models, which are relatively widely used in studies on cerebral infarction.

Previous studies have shown that CD (Sample A of this study) exerts antihypertensive [18,20], antihyperlipidemic [14,16], anti-inflammatory [17,19], and antiapoptotic effects on vascular endothelial cells [15]. Additionally, numerous studies have demonstrated the neuroprotective effects of CD. Ko et al. [36] reported that CD exerted neuroprotective effects by reducing the expression of Bax, a proapoptotic protein, in neuroblastoma 2a cells in a mouse model established through hypoxia–reoxygenation. Kim et al. [37] reported that CD exerted neuroprotective effects by inhibiting the production of reactive oxygen species (ROS) in in vitro and in vivo models of Parkinson′s disease. Nam et al. [38] reported that CD exerted neuroprotective effects by reducing the production of nitric oxide, tumor necrosis factor-alpha, and prostaglandin E2 by activated microglia. Moreover, Cho et al. [21] reported that CD ameliorated cerebral infarction in a tMCAO mouse model of focal cerebral infarction; further, this neuroprotective effect was attributed to the inhibition of microglial activation and neutrophil infiltration. Moreover, Cho et al. [13] performed follow-up magnetic resonance imaging in patients with small-vessel cerebral infarction (SVD), and found that patients who took 600 mg of CD daily for 2 years had a significantly lower recurrence rate of cerebral infarction than patients who took antiplatelet drugs.

In our study, Sample A (CD) allowed a nonsignificant decrease in cerebral infarction, which is inconsistent with the results of a previous study using a rat tMCAO model [21]. This could be attributed to differences between the pMCAO and tMCAO models. Accordingly, 30 mg/kg aspirin did not significantly reduce cerebral infarction, which is inconsistent with the previous report [39,40,41]. In the pMCAO model, since the distal end of the middle cerebral artery is occluded, the resulting cerebral infarction size is relatively small compared with that in the tMCAO model, which involves occlusion of the proximal end. The lack of a significant difference in our study could be attributed to the small sample size.

Cardiotonic pills are used to treat angina, a coronary artery disease, and comprise Salviae Miltiorrhizae Radix, Notoginseng Radix et Rhizoma, and Bomeolum. Over 100 randomized controlled trials (RCTs) have been conducted in China to investigate cardiotonic pills. A systematic review of 60 RCTs reported that cardiotonic pills were more effective than isosorbide dinitrate in treating angina [24]. In vitro studies have attributed the effects of cardiotonic pills to the lowering of endothelial cell–leukocyte adhesion by reducing endothelial cell adhesion molecules, including intercellular adhesion molecule 1 (ICAM-1) and vascular cell adhesion molecule 1 (VCAM-1) [42,43], as well as the inhibition of platelet and leukocyte activation [44]. Zhao et al. [25] reported that cardiotonic pills could effectively prevent myocardial injury and microcirculation disorder after coronary artery ischemia/reperfusion in mice, which was attributed to antioxidant effects by Yang et al. [26]. Additionally, in an animal model of hyperlipidemia, cardiotonic pills exerted antihyperlipidemic effects and improved erythrocyte deformability [45]. Ling et al. [46] reported that cardiotonic pills inhibited arteriosclerosis in a mouse model of high-fat diet-induced atherosclerosis, which was attributed to the inhibition of ICAM-1 expression. Another study described the inhibitory effects of cardiotonic pills on thrombogenesis [47]. A clinical study on patients with hyperlipidemia reported that cardiotonic pills significantly decreased ICAM-1 and E-selectin expression [48].

Although most studies on cardiotonic pills have focused on cardiovascular diseases, cardiotonic pills may be effective in cerebrovascular diseases, since they share the pathophysiological characteristic of arteriosclerosis with cardiovascular diseases. In our study, Sample B + C, which comprised the herbal materials present in cardiotonic pills, significantly reduced cerebral infarction. Therefore, cardiotonic pills may exert neuroprotective effects on cerebral infarction. Accordingly, Lee et al. [27] reported that cardiotonic pills exerted neuroprotective effects against white matter and hippocampal damages induced by occlusion of both common carotid arteries in white mice, which was attributed to inhibition of the activation of inflammatory processes related to microglia and inflammatory mediators. Kwon et al. [29] reported that cardiotonic pills exerted protective effects on neuroglia from oxidative damage, and increased regional cerebral blood flow in white mice. Xu et al. [28] reported that cardiotonic pills reduced the recurrence rate of cerebral infarction/transient cerebral ischemia.

In the MTT assay, compared with the control group, Sample A and Sample B + C did not significantly increase the O.D. value at 48 h after OGD treatment; however, Sample A + B + C (at both the low [A10 + B1 + C0.1 (μg/mL)] and high [A50 + B5 + C0.5 (μg/mL] concentrations) significantly increased the O.D. value. Similar results regarding cell viability were observed in the cell count experiment. These findings indicate a synergistic neuroprotective effect of the Sample A + B + C complex compared with each sample.

Although the pathway by which GCD affected cell viability after OGD was not verified in this study, mitochondrial targeting is presumed to be the mechanism. Mitochondria are essential to maintaining cell energy homeostasis, and thus are inevitably associated with ischemic neuronal cell death [49]. Mitochondria dysfunction is a major factor in ischemia/reperfusion injury that causes neuronal death [50], and mitochondria targeting is being discussed as a neuroprotective strategy for the treatment and prevention of ischemic stroke [51]. A previous study has shown that CD, part of GCD, has a neuroprotective effect by regulating mitochondrial dysfunction caused by ROS generation [34]. Polyphenols, found in natural plants, affect mitochondrial biogenesis by modulating intracellular signaling pathways [52]. The herbal medicines that make up GCD are all natural plants and contain polyphenols, supporting the mitochondrial targeting of GCD. However, this requires clarification through additional experiments.

Compared with the control group, the Sample A + B + C complex extract (fixed concentration of Sample A) and the Sample A + B + C complex extract (fixed Sample B + C concentration) nonsignificantly and significantly decreased the cerebral infarct volume, respectively. Since Sample A and Sample B + C alone did not significantly reduce the cerebral infarction volume, Sample A + B + C could be considered to have a synergistic neuroprotective effect, which is consistent with the results of the OGD experiment. Further, since Sample B + C, but not Sample A, reduced the cerebral infarction volume, Sample B + C could play a greater role in the observed synergistic effect than Sample A. Moreover, since the synergistic effect was observed at a specific mixing ratio, further studies are warranted to determine the optimal mixing ratio.

Although we did not elucidate the mechanism of action underlying the neuroprotective synergistic effects of GCD on acute cerebral infarction, it can be inferred from previous studies on CD and cardiotonic pills. CD and cardiotonic pills have been shown to treat ischemic brain injury through inflammation reduction by inhibiting microglia activation [21,27,38]. Inflammatory processes contribute to acute brain injury after ischemia. Specifically, microglia can be activated to secrete various inflammatory substances and ROS, which further aggravates brain damage. Alternatively, circulating immune cells can be activated, with blood–brain barrier damage after cerebral infarction, promoting the expression of cell adhesion molecules (ICAM-1, VCAM-1) in vascular endothelial cells. As a result, various inflammatory substances are attached, allowing leukocytes to penetrate the brain parenchyma, which secondarily exacerbates inflammation and brain damage [3]. Since the neuroprotective effect of CD could be attributed to the inhibition of leukocyte infiltration [21], and cardiotonic pills can suppress ICAM-1 expression [42,43,46,48], the neuroprotective synergistic effects of GCD could be attributed to the reduction of inflammation through inhibition of the activation of circulating immune cells and microglia.

There has been increasing interest in collateral therapeutics for neuroprotection in cerebral infarction [53,54,55]. Collateral therapeutics delay or prevent damage to the cranial nerves in the ischemic region by enhancing blood flow through the leptomeningeal collateral channel (leptomeningeal or pial collaterals). Currently, proposed approaches to this include increasing circulating blood flow, inducing hypertension, administering vasodilators, temporarily occluding the abdominal aorta, and electrically stimulating the parasympathetic ganglia [54]. Since cardiotonic pills increase local cerebral blood flow [29], the neuroprotective effect of GCD could have resulted from improved collateral circulation. Further studies are warranted to investigate the mechanism of action underlying the efficacy of GCD.

There are several significant aspects of this study. First, a novel alternative for the prevention and treatment of ischemic stroke was presented. CD, which is currently used for the treatment and prevention of ischemic stroke, has a recurrence rate of 6.2% for ischemic stroke due to small vessel disease over 5 years, which is known to be the highest among the previously announced stroke recurrence rates of antiplatelet agents [23]. Since GCD is expected to have better ischemic stroke treatment and prevention effects according to the results of this study, it has potential as an alternative for existing antiplatelets. Second, the combination possibilities and advantages of several herbal medicines were suggested. Western medicines are often used in combination with medications with similar effects, such as dual antiplatelet therapy, to enhance the effect of antiplatelet agents. Considering that combining CD and cardiotonic pills, which have proven neuroprotective effects, leads to better neuroprotective effects of GCD, herbal medicines may have a synergistic effect when medicines with similar effects are combined. Collectively, these processes have opened up the possibility of various applications of herbal medicine as an alternative component for the treatment and prevention of ischemic stroke.

Despite the importance of our findings, future studies concerning physiological parameters of blood samples in the pMCAO model should be conducted. Since changes in the infarct volume in pMCAO may have been influenced by blood parameters at the time, it is necessary to measure physiological parameters by taking blood samples before and after pMCAO. In addition, the excellent neuroprotective effect of GCD was revealed, but the composition ratio of CD and three herbal medicines of cardiotonic pills, Salviae Miltiorrhizae Radix, Notoginseng Radix et Rhizoma, and Bomeolum, could not be concluded. It is essential to search for the optimal composition ratio for the homogenization of medications and maximization of efficacy. Finally, since the models that explored the neuroprotective effect of GCD in this study were in vitro and in vivo, additional verification through future clinical studies is required based on the evidence demonstrated here.

## 4. Materials and Methods

### 4.1. Analysis of Experimental Drugs

#### 4.1.1. Preparation of Experimental Drugs

We obtained Coptidis Rhizoma, Scutellariae Radix, Phellodendri Cortex, Gardeniae Fructus, Rhei Radix et Rhizoma, Salviae Miltiorrhizae Radix, Notoginseng Radix et Rhizoma, and Bomeolum from Kyung Hee Herb Pharm (Wonju, Korea). Samples A, B, and C were prepared as the experimental drugs. Sample A (CD extract) was produced through reflux extraction of a 340 g mixture of Coptidis Rhizoma, Scutellariae Radix, Phellodendri Cortex, Gardeniae Fructus, and Rhei Radix et Rhizoma (4:4:4:4:1) twice, using 2 L of 80% ethanol for 2 h, concentrating the extract, and freeze-drying (yield: 26.5%). The standardization of the Sample A preparation has been previously described [14].

Sample B (Salviae Miltiorrhizae Radix/Notoginseng Radix et Rhizoma root extract) was produced through ethanol (80%) extraction of a 315 g mixture of Salviae Miltiorrhizae Radix and Notoginseng Radix et Rhizoma (ratio, 17.5:3.4) based on the composition ratio of cardiotonic pills in the aforementioned method (yield: 33.3%). Sample C was Bomeolum.

#### 4.1.2. High-Performance Liquid Chromatography (HPLC) Analysis of Samples

HPLC analysis was performed for standardization of Sample B. A BÜCHI Rotavapor R-220 (BÜCHI Labortechnik, Flawil, Switzerland) and deep freezer (IlShin BioBase, Dongducheon, Korea) were used to prepare the raw extraction material. The HPLC system comprised an Alliance 2690 separation module, a Waters 996 photodiode array detector, and the Millenium^32^ Chromatography Manager Version 3.2. Moreover, Nucleosil C_18_ (Waters, USA; 5μm, 4.0 mm × 250 mm I.D.) was used for the column. Salvianolic acid B and tanshinone II A, which were used as reference standards, were purchased from Fluka (Zwijndrecht, The Netherlands). Ginsenoside Rg1 and Rb1 were purchased from Wako (Osaka, Japan). Methanol, acetonitrile, and water, which were the HPLC solvents, were obtained from J.T. Baker (Phillipsburg, NJ, USA). Guaranteed reagents (Sigma, Saint Louis, MO, USA) were used as other solvents for preparing the sample solution.

For the mobile phase of salvianolic acid B and tanshinone II A analysis, we used water (A) and methanol:acetonitrile:formic acid (750:250:1) (B). Additionally, the experiment was conducted using a 280 nm UV detector under the following gradient conditions (0 min: 25% B, 1 min: 40% B, 14 min: 4% B, 22 min 60% B, 23 min 89% B, 40 min 89% B) at a flow rate of 1 mL/min. For the mobile phase of ginsenoside Rg1 and Rb1 analysis, 15% acetonitrile (A) and 80% acetonitrile (B) were used; moreover, the experiment was conducted using a 203 nm UV detector under the following gradient conditions (0 min: 0% B, 20 min: 10% B, 50 min: 30% B) at a flow rate of 1 mL/min.

To analyze salvianolic acid B and tanshinone IIA, 10 mL of 85% methanol was added to 1 g of cardiotonic pills and 100 mg of Sample B, followed by ultrasonic extraction for 60 min and filtration with a 0.45 μm membrane filter to yield the sample solution. To analyze ginsenoside Rg1 and Rb1, 50 mL of 70% methanol was added to 1 g of cardiotonic pills and 100 mg of Sample B, followed by reflux extraction in a water bath for 30 min, filtration and concentration under reduced pressure, and addition of water and ether to the residue to separate a water layer. After adding butanol to the water layer, the butanol layer was separated, concentrated under reduced pressure, dissolved in 5 mL methanol, and filtered with a 0.45 μm membrane filter to yield a sample solution.

### 4.2. Cell Experiments

#### 4.2.1. SH-SY5Y Cell Line Culture

The SH-SY5Y cell line was cultured to identify the neuroprotective effect of the experimental drugs against cell death. The SH-SY5Y cell line is a human-derived neuroblastoma cell line, which proliferates in the presence of serum in the culture solution, with the proliferation stopping and differentiation starting in the serum-free condition. The SH-SY5Y cell line was cultured in a humidified 5% CO_2_ incubator at 37 °C using a culture solution containing 10% fetal bovine serum (FBS, Gibco^®^, Life Technologies, Carlsbad, CA, USA) and 1% antibiotics dissolved in Dulbecco Modified Eagle Medium (DMEM, Gibco^®^, Life Technologies, Carlsbad, CA, USA) and 75 cm^2^; culture flasks. The culture solution was replaced at intervals of 2–3 days; additionally, when the cultured cells reached approximately 70–80% of the dish, the subculture was performed at 10:1. The cultured cells were subcultured at a concentration of 2.0 × 10^5^ cells/mL in 96-well culture plates and Petri dishes, which was exchanged with FBS-free culture solution at 24 h before the oxygen–glucose deprivation (OGD) experiment.

#### 4.2.2. Oxygen–Glucose Deprivation (OGD) and Experimental Drug Treatment

An OGD experimental method was performed to examine the neuroprotective effect of the experimental drugs, as described by Wang et al. [30] and Kim et al. [31], using the cultured SH-SY5Y cell line. For OGD exposure, 95% N_2_/5% CO_2_ without oxygen was injected into the chamber using an anaerobic chamber to induce an anoxic state; further, glucose-free DMEM (Life Technologies, Carlsbad, CA, USA) was used. OGD treatment was performed for 16 h, as described by Wang et al. [30]. After OGD treatment, it was replaced with a normal culture solution (DMEM) containing glucose. After 48 h of OGD exposure, cell viability was identified using the MTT assay, cell counting, and Live/Dead Viability Cytotoxicity Kit.

The experimental drug was simultaneously administered with OGD exposure until examination of the cell viability. The experimental group was divided into seven groups based on the preliminary experimental results: control group, Sample A 10 μg/mL treatment group (A10), Sample A 50 μg/mL treatment group (A50), Sample B 1 μg/mL + Sample C 0.1 μg/mL treatment group (B1 + C0.1), Sample B 5 μg/mL + Sample C 0.5 μg/mL treatment group (B5 + C0.5), Sample A 10 μg/mL + Sample B 1 μg/mL + Sample C 0.1 μg/mL treatment group (A10 + B1 + C0.1), and Sample A 50 μg/mL + Sample B 5 μg/mL + Sample C 0.5 μg/mL treatment (A50 + B5 + C0.5).

#### 4.2.3. MTT Assay

The MTT assay was performed to identify cell viability. Here, 100 µL per well was dispensed and cultured in 96-well plates at a concentration of 2.0 × 10^5^ cells/mL, as described by Carmichael [32]. The test group in which cells were cultured only with the culture solution was used as the control group, while the sample groups were as follows: 10A, 50A, 1B + 0.1C, 5B + 0.5C, 10A + 1B + 0.1C, and 50A + 5B + 0.5C groups. We added 10 µL per well to achieve a final MTT (M5655, Sigma, Saint Louis, MO, USA; stock solution 5mg/mL) concentration of 0.5 mg/mL, followed by incubation for 2 h. After removing the culture solution, 100 µL of dimethylsulfoxide was added and left for 5 min to dissolve the MTT formazan. After 10 min, optical density was measured at 570 nm using an ELISA reader.

#### 4.2.4. Cell Count

In addition to the MTT assay, the cell count method was performed using trypan blue (Sigma, Saint Louis, MO, USA) to identify cell viability. After 48 h of OGD exposure, the cells were washed with PBS and suspended in a PBS solution containing trypan blue to separate live and dead cells, followed by cell counting. Cell viability was expressed as the percentage (%) of live cells (white cells) in the total number of cells.

#### 4.2.5. Live/Dead Viability Cytotoxicity Kit

The Live/Dead Viability Cytotoxicity Kit (Molecular Probes, Eugene, OR, USA) was used to determine the effect of the experimental drugs on cell viability after OGD exposure. This kit contains calcein AM and EthD-1 (ethidium homodimer), which have cell permeability, with live cells showing a green fluorescence (ex./em. ~495 nm/~515 nm). The morphology of the entire cell can be observed. Since EthD-1 cannot penetrate the cell membrane, live cells are stained green by calcein AM. Contrastingly, dead cells with damaged cell membranes are stained bright red (ex./em. ~495 nm/~635 nm) by EthD-1 in the nucleus. After 48 h of OGD exposure, cells were washed with PBS, suspended in PBS solution containing 2 μM calcein AM and 4 μM EthD-1 for approximately 30 min, and stored in a normal incubator. After washing with PBS, fluorescence images were obtained using a fluorescence microscope (Carl Zeiss, Oberkochen, Germany) and a CCD camera (Hamamatsu, Shizuoka, Japan).

### 4.3. Animal Experiments

#### 4.3.1. Animal Preparation

We purchased 5-week-old Institute of Cancer Research (ICR) male mice (20 g), maintained in a sterile state, from Samtako (Seoul, South Korea). The feed for the sterilized laboratory animals was purchased from Purina (Seoul, South Korea), with the mice being allowed ad libitum access to feed and water. The mice were acclimatized for 7 days in a breeding room with a constant temperature (23 ± 1 °C) and humidity (60 ± 10%). All animal experiments were approved by the Institutional Animal Care and Use Committee of Kyung Hee University Medical Center (KHMC-IACUC 11-010, approved on 19 April 2011).

#### 4.3.2. Permanent Middle Cerebral Artery Occlusion (pMCAO) Model

The pMCAO model for focal cerebral ischemia was established as described by Majid et al. [33]. After anesthetizing ICR mice (weight: 25 g–30 g) with 2% isoflurane in a mixed gas of 70% N_2_O and 30% O_2_, a 1.0 cm incision was made between the right eye and ear, followed by opening of the temporal muscle to expose the temporal bone. After identifying the middle cerebral artery under a surgical microscope, a 2.0 mm burr hole was made in the temporal bone using a drill, followed by careful removal of the dura mater. Subsequently, the main trunk at the distal end of the middle cerebral artery was cauterized using a bipolar coagulator; further, the cauterization site was cut using a microscissor to identify the blood flow blockage. After suturing the skin incision site, a body temperature maintenance device (Harvard Apparatus, Holliston, MA, USA) was used to maintain the body temperature at 38 ± 1 °C during the recovery period. The rectal temperature was intraoperatively maintained at 36.5 °C–37.5 °C using a heating pad. We excluded mice that did not turn due to paralysis, or postoperatively developed subarachnoid hemorrhage from the analysis.

#### 4.3.3. Administration of the Experimental Drugs

The experimental drugs were administered at different concentrations to the corresponding mouse group (each group, *n* = 8–10 mice). The number of each group referred to a previous study with a similar protocol [34]. Specifically, to evaluate the neuroprotective effect of Sample A, it was orally administered at 15 mg/kg, 45 mg/kg, and 85 mg/kg immediately and 2 h after pMCAO surgery. For the control group, vehicle and 30 mg/kg aspirin were similarly administered.

To evaluate the neuroprotective effect of Sample B + C, it was orally administered at 2.5 mg/kg + 1.2 mg/kg, 7.5 mg/kg + 1.2 mg/kg, and 22.5 mg/kg + 1.2 mg/kg immediately and 2 h after pMCAO surgery.

To evaluate the neuroprotective effect of Sample A + B + C complex administration, Sample A + B + C complex extracts with different mixing ratios (Sample A was fixed) were orally administered at three concentrations (15:2.5:1.2, 15:7.5:1.2, 15:22.5:1.2 (mg/kg)) immediately and 2 h after pMCAO surgery.

To evaluate the neuroprotective effect of Sample A + B + C complex administration, Sample A + B + C complex extracts with different mixing ratios (Sample B + C was fixed) were orally administered at three concentrations (15 + 7.5 + 1.2, 45 + 7.5 + 1.2, 85 + 7.5 + 1.2 (mg/kg)) immediately and 2 h after pMCAO surgery.

#### 4.3.4. Measurement of Cerebral Infarct Volume

At 24 postoperative hours, the mice underwent cervical dislocation and decapitation; subsequently, the cranial bone was removed around the centrally located sulcus using a rongeur. The excised brain was cut at 2 mm intervals from the third column using a brain matrix slicer. The slices were stained using 1% 2,3,5-triphenyltetrazolium chloride (Sigma, USA) solution at 30 °C for 30 min, and fixed by immersion in 10% neutral-buffered formalin (Sigma, USA). The staining result is the same red for normal brain tissue and white for infarct brain tissue. Because red-stained tissue changes color from light red or pink to dark red in a short period of time (tens of seconds to 1 min) after staining, variations in the time between staining and fixation may result in light and dark concentration differences in the stained red color and do not affect the results of the experiment. Subsequently, the slices were photographed using a digital camera, followed by measurement of the area of cerebral infarction using the ImageJ program (version 1.47). The cerebral infarct volume (mm^3^) was calculated by multiplying the sum of the cerebral infarct areas of all slices by the distance between the slices.

### 4.4. Statistical Analysis

Data are presented as mean ± standard deviation. One-way analysis of variance (ANOVA) or Kruskal–Wallis analysis was performed using GraphPad Prism (version 4); further, Tukey’s multiple comparison test was used for posthoc analysis. Repeated measures ANOVA was used for repeatedly measured data among the groups. Statistical significance was set to *p* < 0.05.

## 5. Conclusions

In the in vitro cerebral infarction OGD model, GCD had a significant neuroprotective effect compared to the control group, but CD did not. In the cerebral infarction in vivo experimental model pMCAO, GCD, and CD had a significant neuroprotective effect compared to the control group. Compared to CD, GCD had a more enhanced neuroprotective effect in acute ischemic stroke, and it was found that there was a synergistic effect by combining Salviae Miltiorrhizae Radix, Notoginseng Radix et Rhizoma, and Bomeolum. These findings indicate that GCD has superior neuroprotective effects compared to conventional treatments, and could be a novel alternative for the treatment and prevention of ischemic stroke.

## Figures and Tables

**Figure 1 pharmaceuticals-16-00596-f001:**
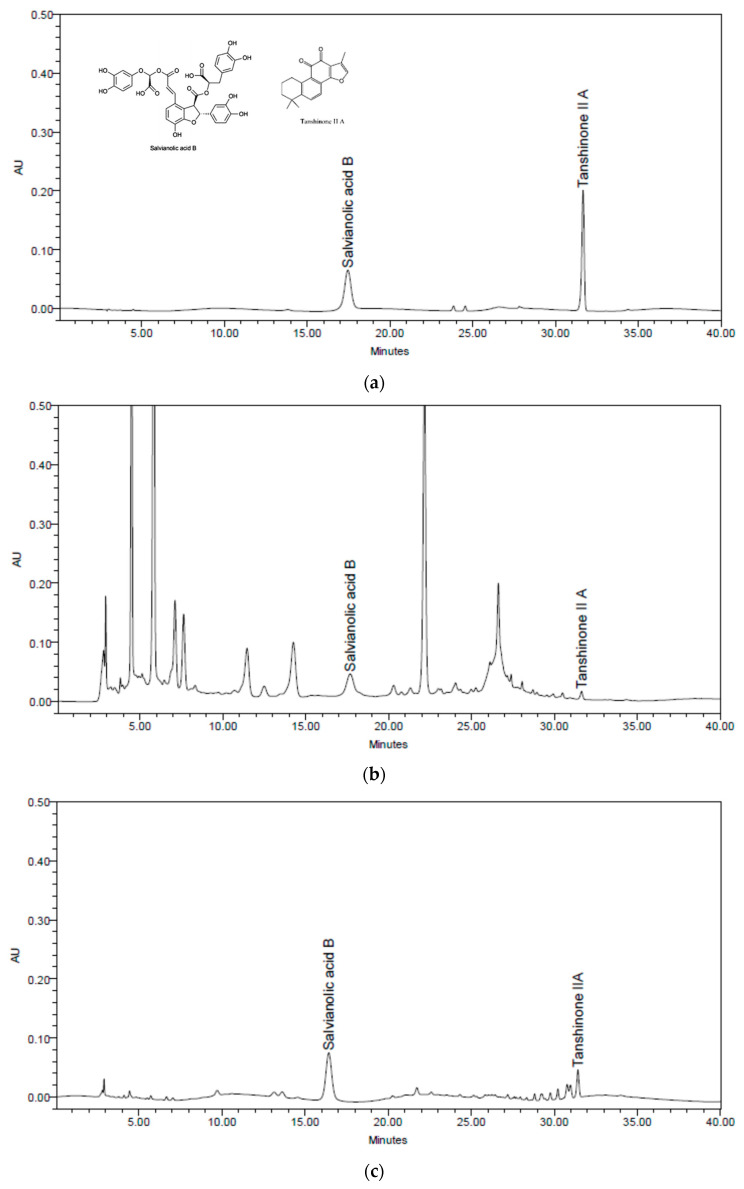
HPLC chromatogram of salvianolic acid B and tanshinone II A. (**a**) A standard mixture of salvianolic acid B, tanshinone II A. (**b**) Cardiotonic pills contained 0.14% of salvianolic acid B and 0.002% of tanshinone II A. (**c**) Sample B contained 2.66% of salvianolic acid B and 0.11% of tanshinone II A.

**Figure 2 pharmaceuticals-16-00596-f002:**
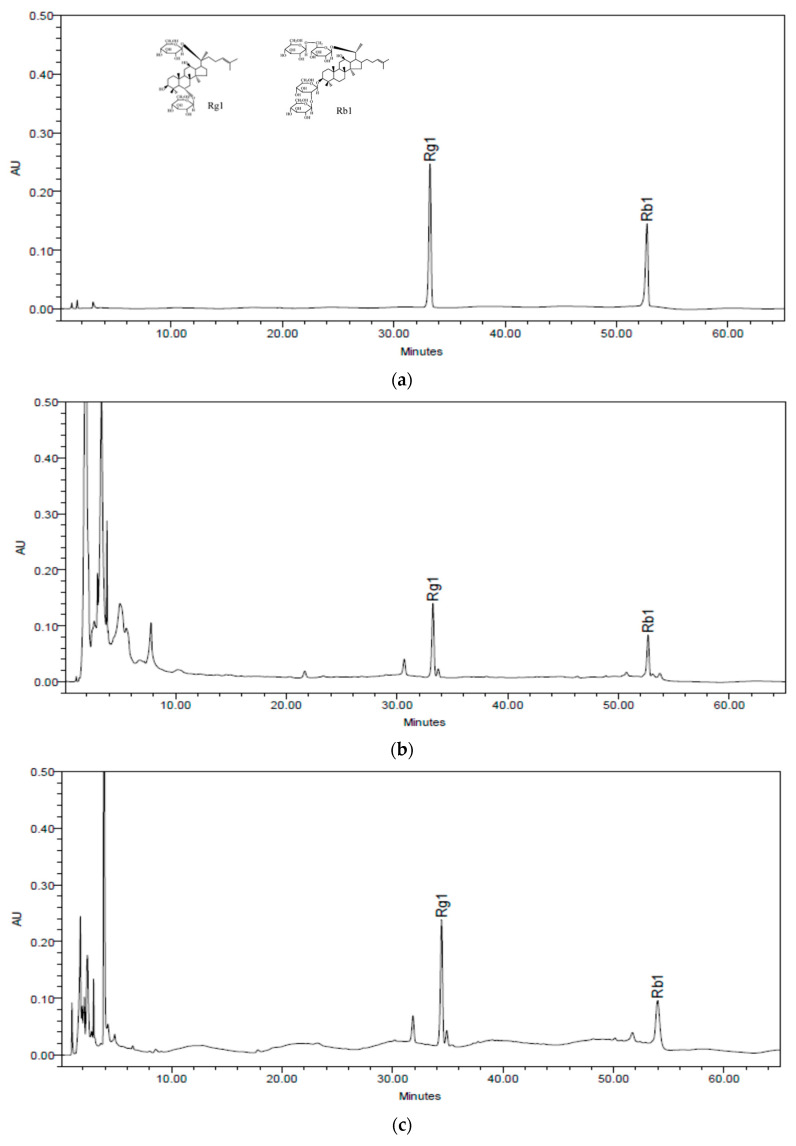
HPLC chromatogram of ginsenoside Rg1 and Rb1. (**a**) Standard mixture of ginsenoside Rg1 and Rb1. (**b**) Cardiotonic pills contained 0.36% of ginsenoside Rg1 and 0.34% of Rb1. (**c**) Sample B contained 0.83% of ginsenoside Rg1 and 0.95% of Rb1.

**Figure 3 pharmaceuticals-16-00596-f003:**
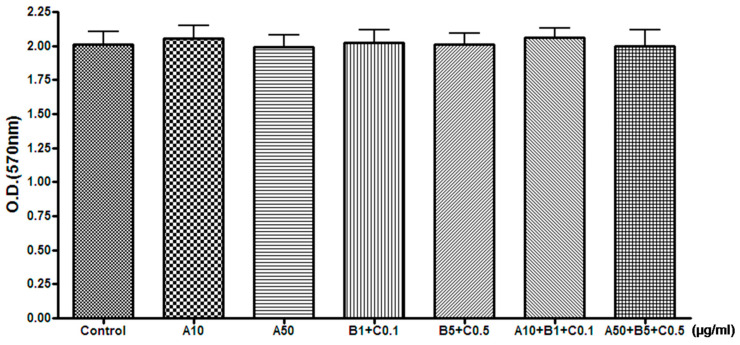
Effects of experimental drugs on the MTT assay results of SH-SY5Y cell viability. (μg/mL, mean ± standard deviation, *n* = 5). No significant difference in the MTT assay of cell viability was observed among groups. A, Chunghyuldan; B, ethanol extract of Radix Salviae Miltiorrhizae and Radix Notoginseng; C, Borneolum Syntheticum; O.D., optical density.

**Figure 4 pharmaceuticals-16-00596-f004:**
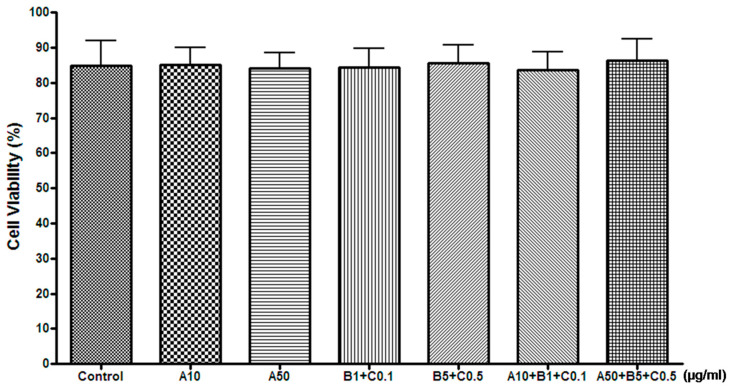
Effects of experimental drugs on SH-SY5Y cell viability by cell count experiment. (μg/mL, mean ± standard deviation, *n* = 5). No significant difference of cell viability was observed among groups. A, Chunghyuldan; B, ethanol extract of Radix Salviae Miltiorrhizae and Radix Notoginseng; C, Borneolum Syntheticum.

**Figure 5 pharmaceuticals-16-00596-f005:**
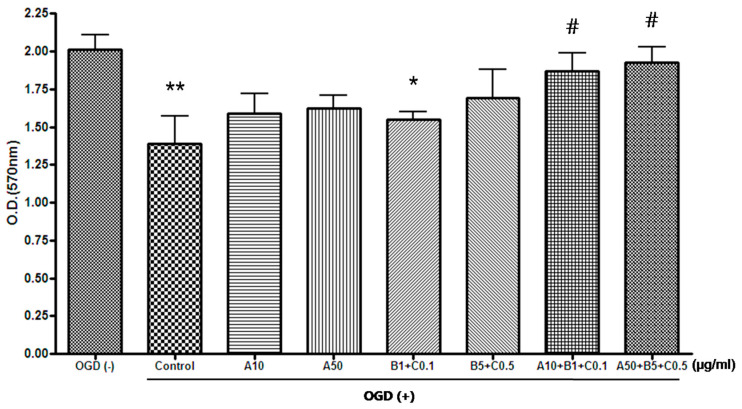
Effects of experimental drugs on MTT assay after OGD in SH-SY5Y cells (μg/mL, mean ± standard deviation, *n* = 5). OGD exposure significantly decreased the O.D. value of the cells. Compared with the control group, Sample A + B + C complex [both concentrations of A10 + B1 + C0.1 (μg/mL) and A50 + B5 + C0.5 (μg/mL)] significantly increased the O.D. value. Sample A and Sample B + C showed an increase in O.D. value compared to control, but there was no statistical significance in all combinations. ** and * indicate *p* < 0.01 and *p* < 0.05 vs. OGD-free control, respectively; ^#^ indicates *p* < 0.05 vs. control by analysis of variance followed by Tukey’s test. OGD, oxygen–glucose deprivation; A, Chunghyuldan; B, ethanol extract of Radix Salviae Miltiorrhizae and Radix Notoginseng; C, Borneolum Syntheticum; O.D., optical density.

**Figure 6 pharmaceuticals-16-00596-f006:**
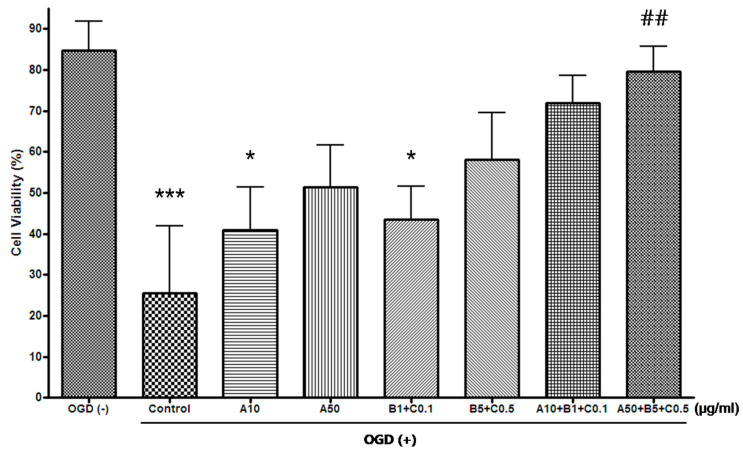
Effects of the experimental drugs on OGD-induced cell death by cell count experiment in SH-SY5Y cells. (μg/mL, mean ± standard deviation, *n* = 5). OGD exposure significantly reduced the number of live cells. Compared with OGD-treated control groups, Sample A + B + C complex (A50 + B5 + C0.5 [μg/mL]) significantly increased the cell viability compared with the control group. Sample A and Sample B + C showed an increase in cell viability compared to control, but there was no statistical significance in all combinations. *** and * indicate *p* < 0.001 and *p* < 0.05 vs. OGD-free control, respectively; ^##^ indicates *p* < 0.01 vs. control by analysis of variance followed by Tukey’s test. OGD, oxygen–glucose deprivation; A, Chunghyuldan; B, ethanol extract of Radix Salviae Miltiorrhizae and Radix Notoginseng; C, Borneolum Syntheticum.

**Figure 7 pharmaceuticals-16-00596-f007:**
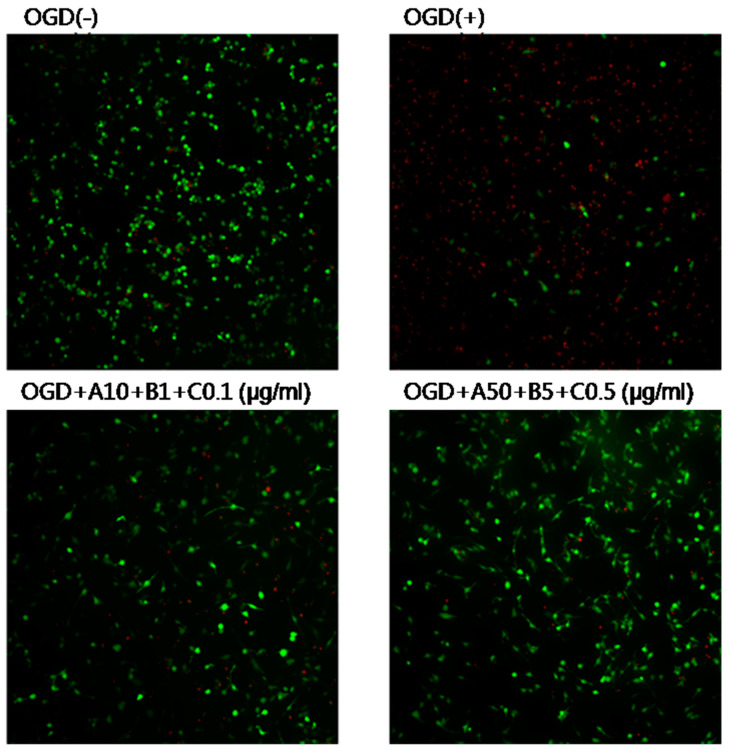
Effects of Sample A + B + C on OGD-induced cell death in SH-SY5Y cells. Green and red dots indicate live and dead cells, respectively. OGD exposure resulted in cell death, which was significantly reduced by treatment with the experimental drugs. OGD, oxygen–glucose deprivation; A, Chunghyuldan; B, ethanol extract of Radix Salviae Miltiorrhizae and Radix Notoginseng; C, Borneolum Syntheticum.

**Figure 8 pharmaceuticals-16-00596-f008:**
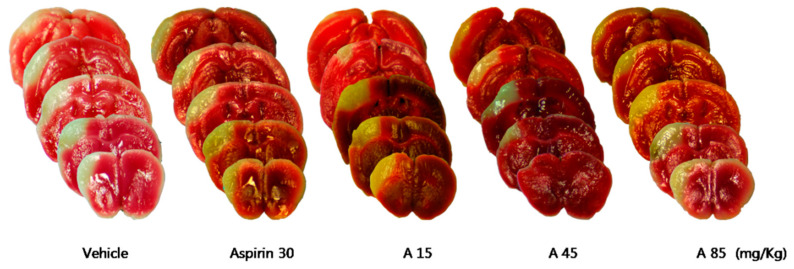
Effects of Sample A on ischemic brain injury after permanent middle cerebral artery occlusion (pMCAO). Infarct volume was measured 24 h after pMCAO. (**Upper**) Representative photographs of infarcted brain slices from vehicle- and Sample A-treated mice. (**Lower**) There was no significant among-group difference in the cortical infarct volume after 24 h (*n* = 8–10 mice for each group). Data represent means ± standard deviation. A, Chunghyuldan.

**Figure 9 pharmaceuticals-16-00596-f009:**
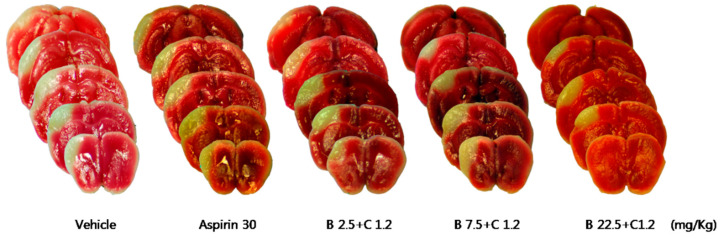
Effects of Sample B + C on ischemic brain injury after permanent middle cerebral artery occlusion (pMCAO). Infarct volume was measured 24 h after pMCAO. (**Upper**) Representative photographs of infarcted brain slices from vehicle- and Sample B + C-treated mice. (**Lower**) Cortical infarct volume was significantly smaller in ‘drug B 22.5 + C 1.2 mg/kg’-treated mice than in vehicle-treated mice after 24 h (*n* = 8–10 mice for each group, vehicle *n* = 19). * Indicates *p* < 0.05 vs. vehicle by analysis of variance followed by Tukey’s test. Data represent means ± standard deviation. B, ethanol extract of Radix Salviae Miltiorrhizae and Radix Notoginseng; C, Borneolum Syntheticum.

**Figure 10 pharmaceuticals-16-00596-f010:**
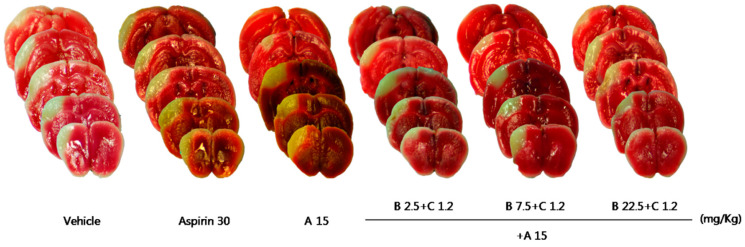
Effects of Sample A + B + C on ischemic brain injury after permanent middle cerebral artery occlusion (pMCAO). Infarct volume was measured 24 h after pMCAO. (**Upper**) Representative photographs of infarcted brain slices from vehicle- and ‘drug A + B + C’-treated mice. (**Lower**) There was no significant among-group difference in the cortical infarct volume after 24 h (*n* = 8–10 mice for each group). Data represent means ± standard deviation. A, Chunghyuldan; B, ethanol extract of Radix Salviae Miltiorrhizae and Radix Notoginseng; C, Borneolum Syntheticum.

**Figure 11 pharmaceuticals-16-00596-f011:**
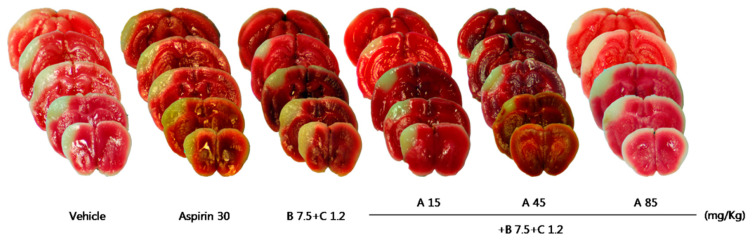
Effects of Sample A + B + C on ischemic brain injury after permanent middle cerebral artery occlusion (pMCAO). Infarct volume was measured 24 h after pMCAO. (**Upper**) Representative photographs of infarcted brain slices from vehicle- and drug A + B + C-treated mice. (**Lower**) Cortical infarct volume was significantly smaller in ′drug A 45 mg/kg + drug B 7.5 mg/kg + C 1.2 mg/kg’-treated mice than in vehicle- and aspirin-treated mice after 24 h (n = 8–10 mice for each group; vehicle, *n* = 8; A 45 + B 7.5 + C 1.2, *n* = 8). Data represent means ± standard deviation. **, *p* < 0.01 vs. vehicle; ^#^, *p* < 0.05 vs. aspirin 30 mg/kg. A, Chunghyuldan; B, ethanol extract of Radix Salviae Miltiorrhizae and Radix Notoginseng; C, Borneolum Syntheticum.

**Table 1 pharmaceuticals-16-00596-t001:** Effects of the experimental drugs on the MTT assay of SH-SY5Y cell viability (*n* = 5).

	Control	Experimental Drug (μg/mL)						*p*
		A10	A50	B1 + C0.1	B5 + C0.5	A10 + B1 + C0.1	A50 + B5 + C0.5	
Mean (O.D.)	2.01	2.06	1.99	2.02	2.01	2.07	2.00	0.900
SD	0.10	0.10	0.09	0.09	0.08	0.07	0.12	

O.D., optical density; SD, standard deviation; A, Chunghyuldan; B, ethanol extract of Radix Salviae Miltiorrhizae and Radix Notoginseng; C, Borneolum Syntheticum.

**Table 2 pharmaceuticals-16-00596-t002:** Effects of the experimental drugs on cell viability by cell count experiment in SH-SY5Y cells (*n* = 5).

	Control	Experimental Drug (μg/mL)						*p*
		A10	A50	B1 + C0.1	B5 + C0.5	A10 + B1 + C0.1	A50 + B5 + C0.5	
Mean (%)	84.8	85.0	84.2	84.3	85.5	83.6	86.3	0.990
SD	7.1	5.0	4.3	5.4	5.2	5.1	6.1	

SD, standard deviation; A, Chunghyuldan; B, ethanol extract of Radix Salviae Miltiorrhizae and Radix Notoginseng; C, Borneolum Syntheticum.

**Table 3 pharmaceuticals-16-00596-t003:** Effects of the experimental drugs on cell viability (MTT assay) after OGD in SH-SY5Y cells (*n* = 5).

	Without OGD	OGD + Experimental Drug (μg/mL)							*p*
		Control	A10	A50	B1 + C0.1	B5 + C0.5	A10 + B1 + C0.1	A50 + B5 + C0.5	
Mean (O.D.)	2.01	1.39 **	1.59	1.62	1.55 *	1.69	1.87 ^#^	1.92 ^#^	0.0001
SD	0.10	0.18	0.13	0.08	0.05	0.19	0.12	0.10	

OGD, oxygen–glucose deprivation; O.D., optical density; SD, standard deviation; A, Chunghyuldan; B, ethanol extract of Radix Salviae Miltiorrhizae and Radix Notoginseng; C, Borneolum Syntheticum; ** *p* < 0.01, * *p* < 0.05 vs. OGD-free control; # *p* < 0.05 vs. control by analysis of variance followed by Tukey’s test.

**Table 4 pharmaceuticals-16-00596-t004:** Effects of the experimental drugs on OGD-induced cell death by cell count experiment in SH-SY5Y cells (*n* = 5).

	Without OGD	OGD + Experimental Drug (μg/mL)							*p*
		Control	A10	A50	B1 + C0.1	B5 + C0.5	A10 + B1 + C0.1	A50 + B5 + C0.5	
Mean (%)	84.8	25.5 ***	40.9 *	51.4	43.5 *	58.2	71.8	79.5 ^##^	< 0.0001
SD	7.1	16.5	10.5	10.2	8.1	11.4	6.7	6.3	

OGD, oxygen–glucose deprivation; SD, standard deviation; A, Chunghyuldan; B, ethanol extract of Radix Salviae Miltiorrhizae and Radix Notoginseng; C, Borneolum Syntheticum; *** *p* < 0.001, * *p* < 0.05 vs. OGD-free control; ^##^ *p* < 0.01 vs. control by analysis of variance followed by Tukey’s test.

**Table 5 pharmaceuticals-16-00596-t005:** Effects of Sample A on infarct volume after pMCAO (*n* = 8–10 mice for each group).

	Vehicle	Experimental Drug (mg/kg)				*p*
		Aspirin 30	A15	A45	A85	
Mean (mm^3^)	30.66	29.09	29.74	25.58	28.25	0.152
SD	5.30	3.78	4.55	3.55	5.55	

SD, standard deviation; A, Chunghyuldan.

**Table 6 pharmaceuticals-16-00596-t006:** Effects of Sample B + C on infarct volume after pMCAO (*n* = 8–10 mice for each group, vehicle for *n* = 19).

	Vehicle	Experimental Drug (mg/kg)				*p*
		Aspirin 30	B2.5 + C1.2	B7.5 + C1.2	B 2.5 + C1.2	
Mean (mm^3^)	30.66	29.09	26.94	26.94	24.49 *	0.030
SD	5.30	3.78	4.89	5.29	4.41	

SD, standard deviation; B, ethanol extract of Radix Salviae Miltiorrhizae and Radix Notoginseng; C, Borneolum Syntheticum; * *p* < 0.05 vs. vehicle by analysis of variance followed by Tukey’s test.

**Table 7 pharmaceuticals-16-00596-t007:** Effects of Sample A + B + C on infarct volume after pMCAO (*n* = 8–10 mice for each group).

	Vehicle	Experimental Drug (mg/kg)					*p*
		Aspirin 30	A15		A15		
				+ B2.5 + C1.2	+ B7.5 + C1.2	+ B22.5 + C1.2	
Mean (mm^3^)	30.66	29.09	29.74	25.74	25.68	26.63	0.081
SD	5.30	3.78	4.55	5.55	3.37	6.50	

SD, standard deviation; A, Chunghyuldan; B, ethanol extract of Radix Salviae Miltiorrhizae and Radix Notoginseng; C, Borneolum Syntheticum.

**Table 8 pharmaceuticals-16-00596-t008:** Effects of Sample A + B + C on infarct volume after pMCAO (*n* = 8–10 mice for each group).

	Vehicle	Experimental Drug (mg/Kg)					*p*
		Aspirin 30	B7.5 + C1.2	B7.5 + C1.2			
				+ A15	+ A45	+ A85	
Mean (mm^3^)	30.66	29.09	26.94	25.68	22.21 **^,#^	25.69	0.003
SD	5.301	3.783	5.287	3.365	4.561	7.454	

SD, standard deviation; A, Chunghyuldan; B, ethanol extract of Radix Salviae Miltiorrhizae and Radix Notoginseng; C, Borneolum Syntheticum; ** *p* < 0.01 vs. vehicle; ^#^ *p* < 0.05 vs. aspirin 30 mg/kg by analysis of variance followed by Tukey’s test.

## Data Availability

Data is contained within the article and Appendix A.

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
