# Peer review of "A Comparative Study on the Neuroprotective Effect of Geopung-Chunghyuldan on In Vitro Oxygen–Glucose Deprivation and In Vivo Permanent Middle Cerebral Artery Occlusion Models"

_pharmaceuticals, 2023, doi:10.3390/ph16040596_

Round 1

Reviewer 1 Report

This manuscript deals with "Neuroprotective effect of Geopung-Chunghyuldan on in vitro oxygen-glucose deprivation and in vivo permanent middle cerebral artery occlusion models". This article claims that using of Geopung-Chunghyuldan could be a suitable for cerebral artery occlusion. Therefore, I suggest a minor correction and require a detailed clarification. Correction to be addressed by the authors as follows: The abstract is not well organized, where the sentences are incomplete and no continuity is there. It would be feasible, if include the significance of the current study in the abstract. A brief description of how the authors selected information from the literature in the databases, as well as what time period they searched for, is missing.
Authors should justify and expand the information on the advantages of this method for biomedical applications.Please discus the role of mitochondria targeting using Geopung-Chunghyuldan in oxygen-glucose deprivation.
Authors should specify the main experimental conditions used on the evidences from the literature. Where they briefly describe the most important data reported in the literature in a homogeneous manner and sequence reinforcing the relevance of Geopung-Chunghyuldan as novel alternative.
Authors should discuss whether the use of Geopung-Chunghyuldan represents a solid alternative to existing therapeutic agents .
Please add below studies to your manuscript in discussion section using below manuscripts:
DOI: 10.1111/j.1742-7843.2010.00612.x

DOI: 10.1155/2021/4946711

Conclusions should reaffirm the fundamental contribution of this paper.

Reviewer 2 Report

Dear Author,

Thanks for submitting your research manuscript entitled "Neuroprotective effect of Geopung-Chunghyuldan on in vitro oxygen-glucose deprivation and in vivo permanent middle cerebral artery occlusion models".

Before giving my final comments as well as the final revision of this manuscript, the author needs to address the following comments scientifically.
Major concerns:

Please find out the following comments:

·         The rationale and purpose behind selecting the SHSY5Y cells in pMCAO model is explained very poorly, irrelevant and incomplete manner throughout the manuscript.

·         Title and abstract is misleading the reader. Title needs to reframe in simply manner.

·         Lack of update as well old & outdated references with incomplete experimental design is another major concern.  

·         In Abstract direct statement “Geopung-Chunghyuldan (GCD), which is a mixture of Chunghyuldan (CD), Radix Salviae Miltiorrhizae, Radix Notoginseng, and Borneolum Syntheticum, is used to treat ischemic stroke in traditional Korean medicine. is confusing. Need to reframe accordingly these types of errors throughout the manuscript.

·         This is preliminary investigation. Reason and explanation behind the updation and initiation of this now as research paper?? With these, lack of limitations in current research, especially, on the basis of current evaluation, it is tough to quote “CD, GCD may allow a more enhanced neuroprotective effect in acute ischemic stroke, indicating a potential synergistic neuroprotective effect. Paper can’t be accepted and encourage to team for further proceeding the methodologies and results.

·         The reviewer found irrational and non-scientific justification in the abstract—introduction and discussion part.

·         Abstract is very poorly written and very confusing. Irrational and fused with repetitions. The reviewer found irrational and non-scientific justification in the abstract—introduction and discussion part.

-          The author fails to explain the relevant justification in the introduction as mentioned in the discussion part.

·         A major drawback is a lack of supporting pre-clinical and clinical evidence regarding targeting drugs.

·         Throughout the manuscript, the main focus is not clear. Complete mismatch of abstract, introduction, results and discussion.

Title:

·         Mismatch of title with relevant introduction and conclusive remarks in the conclusion part.

Abstract:

-     The rationale behind this research is not well explained, and several major concerns still constrain the reviewer's enthusiasm for publishing this manuscript.
Introduction:

- The basic literature is not well written and does not even include any literature on alternative approaches with updated references regarding involvement of current drug treatment/techniques used in pathogenesis and development of neurological illnesses especially pMCAO.

- Authors fail to justify the correlation, and almost irrational and common information is present in the introduction part.

Material and methods:

-     Major drawback is the lack of supporting references and incomplete experimental and paradigms.

- All biochemical parameters are poorly explained without any sense, without any references.

- Provide all biochemicals kits (in-vitro assay) numbers along with their city, country in all individual parameters in all expressions, blots, etc.

- In order to support the assessment of all mentioned parameters in his study, the author should provide all the source documents and data he/she has followed for all assays and estimates.

- How was the dosing determined? Dose-responses should be performed.

- How was the sample size determined? Ideally, a priori sample size calculation should be performed to determine the appropriate sample size.
- Normality and variance homogeneity should be assessed across all groups of the same outcome variable and not individual experimental groups. If the data were not normally distributed or variance homogeneity was not met, nonparametric tests need to be performed.
Parametric data should be reported as mean +/- SD, while nonparametric data should be given/displayed as median and interquartile range. Longitudinal data should be analyzed using repeated measures tests.

Results:

-          All results are very-very poorly explained. Revised all.

-          Due to lack of Statistical Legend description, all figures (Figure 5 & 6) are not acceptable in current form; simplify in proper manner.

-          Quantification graphs of figure 8,10, and 11, stat is not properly mentioned. Re-check and edit accordingly.

-          Image (Figure 7) is highly blurred, without scaling and there is no clarity for easy understanding. Not acceptable in current form. Need to provide high resolution images?

-          Re-check stat of figure 5 and 6, and confirm either statistical symbol are properly mentioned in graphs or not? I am sure, symbols are wrongly or manipulated???

-          Stat is another major concern. Need to verify. Therefore, provide the supplementary data of all graphs for further verification. Without this, article can’t proceed further.

-          Results need more clarification and significant justification. Differentiating between the outcome and the discussion sections is quite difficult.

-          High note: Must provide all results description and Use proper statistical reporting: i.e. for the results of each statistical test, the authors should report the statistical test that was applied, the test statistic (e.g. t, U, F, r), degrees of freedom as subscripts to the test statistic, and the exact probability value, including those for normality and variance homogeneity tests. Statistics should be reported in APA format, i.e.: t(df) = value, p = value; F(df1,df2) = value, p = value; r(df) = value, p = value; [chi]2 (df, N = value) = value, p = value; Z = value, p = value.  Include statements on the tests for normality and variance heterogeneity and respective results. If the data were not normally distributed or variance heterogeneity was not met, nonparametric tests need to be applied.

Discussion:

-     To address the outcome of in-vivo measures/results separately avoiding the ischemic stroke associated neurocomplications and maintaining physiological condition and how they correlate with the existing literature, it would be better if the author restructured to take a more critical approach for effective in neurotoxic conditions.

- The explanation of all is very poor, and need to specify the scale bar properly.
-     In the discussion and the conclusion, the aims, rationale, and future perspectives are not evident clearly in relation with in-vitro and in-vivo experimentation.
-     The discussion is usually unorganized at the beginning to address all the observations and evaluate them at the end. It makes the results easier to contextualize and simpler to comprehend.

- Furthermore, a minimal critical analysis should be provided, along with current study limitations as well the future perspective as separate paragraph.

Conclusion:

-          Need to revise the conclusion in a scientific manner. Not accepted in its current form.

-          This reviewer considers that this paper cannot be published in the present form. A detailed revision shortening, ordering and following the commented ideas could improve this interesting paper in a significant manner.

-          Several typewriting mistakes are present and needing correction. This reviewer remains at entire disposal for the next version.

Reviewer 3 Report

This work evaluated the synergistic neuroprotective effects of Geopung-Chunghyuldan (GCD) in combination with Salviae Miltiorrhizae Radix, Notoginseng Radix Et Rhizoma, Bomeo-lum using oxygen-glucose deprived (OGD) SH-SY5Y cells and a mouse model of stroke with permanent occlusion of the middle cerebral artery (pMCAO). The authors used in vitro and in vivo cerebral infarction models for experiments of cerebral ischemia. This reviewer is concerned about the following points.

1In the pMCAO model, changes in physiological parameters in blood samples before and after cerebral ischemia are unknown.

2 It is unclear whether the cerebral infarct lesions in the Vehicle pMCAO model reproduce the infarct lesions in the drug-treated group.

3 This reviewer questioned the appropriateness of aspirin as a positive control in the pMCAO model.

4 Although the neuroprotective effect of the combination of CD and inotropic agents appears likely, conclusive evidence is lacking because the content and combination ratio of cerebral ischemia Geopung-Chunghyuldan (GCD) and Salviae Miltiorrhizae Radix, Notoginseng Radix Et Rhizoma, Bomeo-lum remains unknown.

Reviewer 4 Report

The authors of the manuscript investigated Geopung-Chunghyuldan (GCD), a traditional Korean medicine, for its effects on the treatment of ischemic stroke in comparison with another traditional medicine Chunghyuldan (CD). The in vitro and in vivo results indicated that GCD may allow a more enhanced neuroprotective effect in acute ischemic stroke, compared to CD. The conclusion does not sound very robust due to the small number of tested samples (n=8-10 mice used in the pMCAO model, and statistically significance with ** is not very high from the control), but the work may still provide some valuable insights for scientific society. So, this reviewer would like to recommend its publication after addressing the following points:

 (1)    Test Rg1 and Rb1 as well as salvianolic acid B and tanshinone IIA in the extraction from the most effective drug mix A45+B7.5+C1.2 by HPLC to see what is different from these in CD and use the results to support their conclusions.

(2)    Add number ‘n’ of the tested samples for each Table, like what they did for each figure in the manuscript.

(3)    In Figure 11, clearly indicate how many mice were used in vehicle and drug A 45mg/Kg + drug B 7.5mg/Kg + C 1.2mg/Kg, respectively.

Round 2

Reviewer 2 Report

Dear Author, 

After careful revision, manuscript can be accepted for further publication. 

Author Response

The reviewer didn't point out anything. 

Reviewer 3 Report

The authors carefully responded to the reviewers' suggestions and the manuscript was revised in response. The revised manuscript appears to have reached an acceptable level.

Author Response

The reviewer didn't point out anything. 

Reviewer 4 Report

According to the authors’ response, the authors of this manuscript addressed some of this reviewer’s comments but stated that it would be difficult to explore the mechanism underlining the conclusions (which were not very robust in experimental as displayed in the manuscript). It is generally true for traditional medicine from natural products. To encourage a young scientist and help his career development, this reviewer would recommend his publication after addressing other reviewers’ comments.

Author Response

The reviewer didn't point out anything.